# A Study on Model of Psychotherapy Narration Focused on Mental Well-Being for Stress Management in the Elderly

Eunyoung Kim [1,2], Seongjoon Kim [1,3] and Jongtae Rhee [1,*]

1   Department of Industrial and Systems Engineering, Dongguk University-Seoul, 30 Pildong-ro 1-gil, Jung-gu, Seoul 04620, Republic of Korea

2   Department of Music Therapy, Graduate School, Sookmyung Women's University, 100 Cheongpa-ro 47-gil, Yongsan-gu, Seoul 04310, Republic of Korea

3   Department of Arts Therapy, Education and Counseling, Hanyang University-Seoul, 222 Wangsimni-ro, Seongdong-gu, Seoul 04763, Republic of Korea

*   Correspondence: jtrhee@dongguk.edu

**Abstract:** Psychological well-being is vitally important for the quality of life of the elderly and is only increasing in importance with the rapidly increasing elderly population worldwide. Emerging elderly problems include a deterioration in physical function, loss of friends or spouse, reduced social participation, and reduced economic ability. Hence, the importance of coping with and managing stress in the elderly is also rapidly increasing. This study proposed psychotherapy narration was designed to assist elderly mental well-being by combining person-centered therapy, positive psychology, and cognitive behavioral therapy. Extending from current mainstream psychotherapeutic methods, postmodern psychotherapeutic techniques based on various psychological theories or techniques have begun to be more widely applied. However, almost no previous studies have developed a systematic psychotherapy narration for the elderly. Therefore, this study developed a postmodern psychotherapeutic narration and confirmed its aspects by analyzing elderly satisfaction regarding the corresponding emotion. This satisfaction analysis study found the value of the psychotherapy-narrative model according to the elderly's stressful situations and emotions. This study can be an initial model of postmodernist-psychotherapy narration for the elderly. Therefore, based on the model of this study, future-oriented development and research on the diversity of the elderly and the effects of each narration are important. The future of this study will give mental self-sustainability to clients who need psychotherapy.

**Keywords:** psychotherapy; narration; mental health; well-being; senior; elderly stress

## 1. Introduction

Social interest in a happy life for the elderly has increased with the rapidly increasing elderly population worldwide, and elderly problems have emerged as serious social problems. Elderly problems include social problems due to rapid social change [1]. The elderly generally experience physical and cognitive decline, loss of friends or spouses, reduced social participation, and reduced economic ability [2]. Barer and Johnson [3] suggested that these represent social problems that are difficult for the elderly to cope with alone, leading to increased disease levels, loss of sensory function, economic difficulties, loneliness, memory loss, and social isolation. According to the statistics of the elderly by the National Statistical Office (2023), health problems (65.2%) and economic problems (53.0%) were among the difficulties of the elderly over 65 years of age [4]. Thus, the South Korean elderly are experiencing considerable stress in their daily lives due to various problems, including loss of economic power due to retirement, psychological loss, and deteriorating physical health.

Situations where the elderly experience stress can be broadly divided into four categories: physical, economic, social, and mental stress [5]. Most elderly experience physical

difficulties due to aging in the developmental stage, and the most common cause of elderly stress is due to physical changes [6], with many of the elderly experiencing maladjustment or difficulty accepting psychological and physical changes. They often lose social status and societal roles due to their deteriorating health, creating a vicious cycle leading to significant psychological problems. Elderly economic difficulties due to rapidly changing social structures in modern society include poverty [7] (as the elderly generally lose their societal roles and suffer from sharply reduced incomes after the age of 65). These economic difficulties pose a significant threat to their quality of life, including basic maintenance costs, leisure, interpersonal relationships, and health maintenance [8].

Social relationship elements generally reduce as elderly activity levels decrease [9], leading to difficulties with interpersonal relationships or interactions with others, and eventually psychological atrophy. This reduced social activity also makes it difficult to maintain a healthy mind and body and eventually impacts their overall quality of life. Elderly stress is often accompanied by negative emotions, including clinical depression, anxiety, and general dissatisfaction with their lives. Cognitive functions gradually deteriorate, including attention span, memory, reasoning, learning, and problem-solving ability [10], causing problems communicating with others and maintaining social relationships. These issues strongly contribute to elderly stress and eventually make it difficult to lead normal lives.

Many previous studies have confirmed that the elderly experience various and complex difficulties due to these physical, economic, social, and psychological aspects. These generally manifest as continuous problems with complex interconnectedness. Elderly problems have previously been considered personal or family problems, but this attitude is rapidly changing as we enter an aging society, with more appreciation that these are not individual problems, but rather problems that all society members should deal with together. The importance of coping with and managing stress to improve the quality of life of the elderly is increasing, and the need for emotional management caused by this stress has also been raised. Therefore, effective programs to support the elderly, e.g., counseling, etc., must carefully consider personal circumstances.

Various indicators and scales have been proposed to measure elderly problems. This study employed the comprehensive elderly problem scale (CEPS) [11], a modified form of the aging problem scale (APS) [3], based on the same four factors as the APS: health, social, emotional, and economic. However, CEPS also considers various elderly emotions arising from the main factors. Kim and Min [12] showed that sorrow, disappointment, anger, anxiety, pleasure, and tranquility impacted elderly experiences, whereas Kim et al. [13] found joy, sadness, anger, surprise, fear, and tranquility to be significant emotions for the quality of life of elderly Koreans.

Properly managing stress and emotions is important for the psychological well-being of the elderly. Mental well-being helps with psychological well-being and protects from chronic or acute stress [14], is an important factor in various mental disorders, and is closely related to therapeutic effects. Although many emotional disorders such as major depressive disorder, panic disorder, social phobia, and obsessive-compulsive disorder have been somewhat successfully treated through cognitive behavioral or drug therapies, patients often have residual symptoms. Well-being treatment for patients with residual symptoms increases treatment effects, reduces residual symptoms, and helps maintain patient psychological stability for a longer period compared with only cognitive behavioral therapy [15]. Therefore, the mental well-being of the elderly is an important factor in their mental health, helping not only with stress management but also with alleviating emotional disorders.

Psychological counseling has been proposed as important to help the elderly manage stress and psychological emotions. However, it is difficult for most elderly to visit hospitals to receive psychological counseling due to being unaware of the need for counseling, pandemic situations, economic difficulties, and social fear [16]. Consequently, various mobile applications, videos, and phones have been provided for remote rather than in-

person counseling [17], which has been widely shown to be effective, particularly in pandemic situations, such as the COVID-19 pandemic.

However, research regarding remote emotional management methods that can be systematically and self-resolved for the elderly remains rudimentary. This study developed a suitable psychotherapy narration for mental well-being in the elderly and analyzed its satisfaction. The proposed psychotherapy narration was developed so the client could use it remotely at any time to consider or discuss recent counseling environments and included diverse responses to each emotional state according to the elderly individual's stressful situations. The psychotherapy-narration model considering the various situations of the elderly can help the mental well-being of the elderly. Our research questions are as follows:

RQ1. What is the psychotherapy-narration model focusing on mental well-being for stress management in the elderly?

RQ2. What is the level of satisfaction (by specific situation and emotion) with the psychotherapy-narration model focusing on mental well-being for stress management in the elderly?

Following previous studies, we selected four areas as major stress situations in CEPS (health, social, emotional, and economic) along with six emotions (sadness, anger, fear, surprise, joy, and tranquility) to produce twenty-four distinct psychotherapy narrations. The proposed psychotherapy narration was designed to assist the elderly with their mental well-being by combining person-centered therapy (PCT), positive psychology, and cognitive behavioral therapy (CBT).

## 2. Psychotherapy Narration

As discussed in Section 1, elderly experience stress in various situations, and their mental well-being is important for appropriate emotional expression and acceptance. However, opportunities for face-to-face programs and counseling for elderly stress management can be limited due to current situations, such as COVID-19. In addition, the Korean elderly are quite reluctant to undertake counseling or psychotherapy. This non-face-to-face type of psychotherapy can reduce the psychological burden of the client and can be conveniently implemented [18]. Therefore, this study developed a psychotherapy narration for elderly mental well-being considering various elderly relevant situations that can be used in remote interactions. The proposed narration was developed to cope with various elderly situations and emotions by applying various psychological theories, as summarized in Table 1.

**Table 1.** Psychological theories embodied in the proposed psychological narration.

| Psychotherapy Type | Code | | Treatments |
|---|---|---|---|
| Person-centered theory (PCT) | 0 | | Narrator attitude |
| Positive psychology | 1 | 1 | Positive-psychological capital |
| | | 2 | Quality of life coaching |
| | | 3 | Well-being therapy (WBT) |
| Cognitive behavioral therapy (CBT) | 2 | 1 | Relaxation techniques |
| | | 2 | Cognitive behavior coaching |
| | | 3 | Exposure therapy |
| Acceptance and commitment therapy (ACT) | 3 | 1 | Acceptance |
| | | 2 | Cognitive defusion |
| | | 3 | Contact with the present moment |
| | | 4 | Values |
| Mindfulness-based stress reduction program (MBSR) | 4 | 1 | Watching the breath |
| | | 2 | Body scan |

### 2.1. Person-Centered-Theory

Psychology encapsulates three main approaches: psychoanalysis, behaviorism, and humanism. Humanism encapsulates humanistic approaches and was previously called

human-centered theory, whereas modern psychoanalysis focuses on the human unconscious, and behaviorism argues that personality is formed under influences from environmental factors.

Humanism emphasizes positive human aspects, such as love, creation, value, meaning, self-realization, and the relationship between the counselor and the client. Rogers is widely regarded as the founder of modern humanistic psychology, together with Maslow [19]. Rogers' human-centered theory is based on the counselor's positive acceptance, sincere understanding, and empathy, rather than their theoretical knowledge and subjective thoughts (arguing that through this, all human beings can have positive experiences in a relationship) [20]. This process helps clients grow into fully functioning persons, i.e., creative persons, by helping them believe in their potential. Corey [21] argued it is important for the counselor to help improve their clients' self-awareness in-person-centered theory, with a particular focus on the here and now.

Counselor attitudes in the proposed narration were developed based on this person-centered theory. Thus, counselors prepared an environment for their clients' psychological well-being by harmonizing with other psychological techniques encapsulating the core attitudes of concordance, sincerity, and empathetic understanding.

*2.2. Positive Psychology*

Psychology has three missions: treat mental disorders, discover and nurture outstanding talents, and help everyone to be happy. However, most studies prior to the second half of the 20th century focused on negative human aspects, such as treating mental disorders [22], with studies considering negative aspects being 17-fold more than those considering positive aspects, which significantly reduced and limited positive human aspects [23]. Therefore, the proposed narration applied many positive-psychological elements regarding positive elderly aspects and mental well-being since these are highly related to quality of life. Seligman [24] founded positive psychology, presenting conditions for a happy life by considering three aspects: a 'pleasant life' to live with positive emotions about the past, present, and future; an 'active life' to discover and use representative strengths; and a 'meaningful life' to discover and give valuable meaning to people, life and actions. Kim [25] confirmed that psychotherapy based on positive psychology effectively reduced sadness and depression in the elderly and alleviated their negative emotions. Ultimately, it can change their quality of life by giving their life meaning and value.

The proposed narration applied three positive psychology theories or techniques for elderly mental well-being: positive-psychological capital, quality-of-life treatment, and well-being therapy. Positive-psychological capital focuses on individual strengths, happiness, and potential rather than individual problems or weaknesses [26] and considers resilience, i.e., the ability to withstand frustrated situations and return to the original state when faced with difficult problems or situations.

Frisch [27,28] developed the positive psychology-oriented quality-of-life treatment. CASIO is a representative model of quality-of-life treatment, evaluating overall satisfaction based on causal relationships between circumstances, attitudes, standards, and importance. Quality-of-life therapy focuses on encouraging clients to understand and apply positive aspects of themselves to improve their happiness and strengths. The treatment improves the overall quality of life and improves emotional stability and intimacy with others in depressed patients [29].

Fava [30] developed well-being therapy (also called positive psychotherapy) based on the Ryff psychological well-being model. Well-being therapy aims to improve control of situations, positive relationships with others, autonomy, personal growth, purpose in life, and self-acceptance. The focus is to increase clients' awareness of their well-being experience by recording well-being moments and emotions to recall. The treatment is based on studies that showed a significant preventive function, protecting individuals from chronic or acute stress and hence increasing psychological well-being [14]. Various

positive psychology theories and techniques are appropriate and effective for elderly mental well-being and can be more effective when combined with cognitive behavioral therapy.

### 2.3. Cognitive Behavioral Therapy

Cognitive behavioral theories argue that an individual's emotions and behaviors are influenced by cognitive processes, and hence emotions and behaviors can be changed positively by modifying these cognitive processes. Ellis [31] and Beck [32] demonstrated and developed CBT effectiveness by combining cognitive and behavioral therapy. CBT is goal-oriented and solution-focused, i.e., focusing on current behavior rather than past experiences. The overall process is for clients to discover negative thoughts and beliefs through counseling and train themselves to correct them, thereby enabling clients to have rational emotions and thoughts.

Engels and Verney [33], and Ken [34] confirmed CBT's effectiveness for the elderly by reviewing several studies related to depression and life satisfaction. Thus, CBT counseling is a solution-oriented therapy that seeks to change the client's thoughts and beliefs by focusing on current situations. The proposed narration applied relaxation techniques, cognitive behavior coaching, and exposure therapy in cognitive behavioral therapy techniques for elderly mental well-being.

Relaxation techniques are part of behavior therapy, including imagery, gradual muscle relaxation, and breathing control. Repetitive training in relaxation techniques can reduce or eliminate anxiety arising from stressful situations. The elderly are less active, and muscle tension can be doubled by stress if the body is in a stiff state [35]. Therefore, relaxation techniques can be an effective coping method for the elderly in stressful situations. Cognitive behavior coaching is where the counselor understands the client's thoughts and beliefs and changes them positively so they can act more productively [36]. This is customized coaching focusing on client well-being and achievement by applying CBT techniques. Counselors use integrated approaches combining cognitive, behavioral, and imaginative problem-solving strategies [37]. Exposure therapy helps clients who are anxious in a stressful situation to face the situation strategically rather than avoid it. Safety behavior to relieve client anxiety provides immediate, temporary relief, but eventually reinforces the avoidance behavior and perpetuates client problems [38]. Therefore, the therapist provides a strong therapeutic rationale for the client to realize the necessity of facing the problem, gradually increasing the client's courage to face the situation. The process is a general method of exposure therapy [39] that helps reduce or alleviate elderly anxiety in various stressful situations.

Previous studies have proposed various CBT techniques to help construct rational thinking for clients' psychological well-being. The proposed treatment is an effective method to guide elderly life and can generate synergistic effects in combination with other treatments.

### 2.4. Acceptance and Commitment Therapy

Hayes [40] proposed acceptance and commitment therapy (ACT), derived from CBT. ACT takes the universal view that human suffering is a normal process and helps the client to have the psychological flexibility to accept negative emotions. The overall goal is to disrupt cognitive fusion and avoidance of experiences, helping clients to move toward the values they desire by means of various methods related to their internal events [41]. The focus is to provide clients with the ability to engage with here-and-now experiences and subsequently encourage them to find worthwhile goals. Psychological flexibility is the ability to continue to maintain behavioral changes selected during treatment [42].

The proposed psychological narration applies acceptance, cognitive defusion, contact with the present moment, and ACT element values for elderly mental well-being. The four theoretical processes are fluidly interrelated [41].

First, acceptance is the willingness to experience existing situations rather than experiential avoidance of negative situations encountered in life.

Second, cognitive defusion means weakening and separating the association and function of emotion and language. Distancing yourself from your thoughts helps you respond more objectively and uncritically to negative situations.

Third, contact with the present moment stimulates internal and external stimuli by focusing on experiences and stimuli in the here and now. Being aware of these inner experiences reduces negative stimuli from cognition.

Fourth, values pursue the meaning and direction of one's life without cost or intervention in that direction.

Shin [43] showed that ACT could effectively change elderly anxiety and psychological rigidity for those experiencing chronic pain, and Kim [44] showed that ACT improved self-esteem and self-integrity levels, which represent the overall quality of life of the elderly, and effectively reduced depression. Thus, ACT helps clients develop a true acceptance attitude, rather than avoidance of experience, and assists the client in developing an objective perspective. Consequently, ACT treatment can be a cognitive basis for elderly mental well-being and can be used in multiple and harmonious ways with other treatments.

### 2.5. Mindfulness-Based Stress Reduction

Kabat-Zinn developed mindfulness-based stress reduction (MBSR) meditation based on practices passed down by monks who inherited the teachings from Jinul masters from 12th-century Korean Buddhism [45]. MBSR is the awareness that comes from paying attention in a particular way. This rising awareness regulates personal energy and attention, influences and changes our experiences, allows the full experience of all areas of humanity, and facilitates experiencing the full range of our relationships with others and the world. Lee [46] argued that the main MBSR function was to pay attention to one's own breathing and simple sounds to maintain the balance between body and mind. This process can harmonize the mind and body and relieve tension. Kabat-Zinn [47] subsequently developed the popular MBSR meditation program, currently practiced by more than 500 companies selected by Fortune magazine and 700 medical institutions worldwide. Korea is implementing MBSR as one of the psychotherapeutic techniques to control emotional pain and maladaptive behavior [48].

For the proposed psychological narration applies observation of the breath and body scanning as MBSR techniques for the mental well-being of the elderly. Watching the breath, also called sitting meditation, comprises meditation where you sit on the floor with your back straight and pay attention to your breathing. The body scan comprises meditation, where you pay attention to each part of your body and observe changes in sensations and emotions. These meditation methods have been shown to have therapeutic effects on anxiety disorders, depression, and stress management and are becoming more commonly used for physical and mental diseases in university hospitals across the US and Europe [46]. Previous studies have also shown that MBSR therapy provides clients with the opportunity to recognize themselves calmly through their own breathing and whole-body senses. The treatment is appropriate to reduce stress for the mental well-being of the elderly and can be more effective when used in combination with other treatments.

## 3. Methods

### 3.1. Study Design (Proposed Psychological Narrative)

Figures 1 and 2 summarize the various psychological theories and techniques discussed in Sections 2.1–2.5 included in the proposed psychological narrative. Twenty-four psychotherapy narratives were developed by considering six emotion types and four stress situations for the elderly. These were subsequently expanded to 72 narrations by dividing emotional intensity on a 3-point scale: high, medium, and low.

**Figure 1 — Physical**

| Physical | Level | \ | \ | \ | \ | \ |
|---|---|---|---|---|---|---|
| Sadness | H | 0 | | 2-1 | 3-1 | 4-2 |
| Sadness | M | 0 | 1-3 | | 3-3 | |
| Sadness | L | 0 | | | | 4-2 |
| Anger | H | 0 | | | 3-2 | 4-1 |
| Anger | M | 0 | 1-3 | | | |
| Anger | L | 0 | | | 3-3 | 4-2 |
| Fear | H | 0 | 1-3 | 2-1 | | |
| Fear | M | 0 | 1-2 | | | |
| Fear | L | 0 | | | | 4-2 |
| Surprise | H | 0 | | 2-1 | | 4-1 |
| Surprise | M | 0 | | 2-2 | | |
| Surprise | L | 0 | 1-1 | | | |
| Joy | H | 0 | 1-1 | | | |
| Joy | M | 0 | 1-1 | | 3-3 | |
| Joy | L | 0 | 1-3 | | | |
| Tranquility | H | 0 | 1-2 | | | |
| Tranquility | M | 0 | 1-2 | | | |
| Tranquility | L | 0 | 1-3 | | | |

**Figure 1 — Economic**

| Economic | Level | \ | \ | \ | \ | \ |
|---|---|---|---|---|---|---|
| Sadness | H | 0 | | 2-2 | 3-1 | |
| Sadness | M | 0 | 1-2 | | 3-4 | |
| Sadness | L | 0 | 1-2 | | | |
| Anger | H | 0 | | 2-2 | 3-2 | |
| Anger | M | 0 | | 2-2 | 3-4 | |
| Anger | L | 0 | | 2-1 | | |
| Fear | H | 0 | 1-2 | | 3-2 | |
| Fear | M | 0 | | 2-2 | 3-4 | |
| Fear | L | 0 | | 2-2 | | |
| Surprise | H | 0 | 1-3 | | 3-4 | |
| Surprise | M | 0 | 1-1 | | 3-4 | |
| Surprise | L | 0 | | | 3-1 | |
| Joy | H | 0 | 1-3 | | | |
| Joy | M | 0 | 1-3 | | | |
| Joy | L | 0 | 1-1 | | | |
| Tranquility | H | 0 | 1-1 | | 3-4 | |
| Tranquility | M | 0 | | | 3-1 | |
| Tranquility | L | 0 | 1-1 | | | |

**Figure 1.** Psychological theories and techniques included in the proposed psychological narrative. H = high, M = medium, L = low.

**Figure 2 — Mental**

| Mental | Level | \ | \ | \ | \ | \ |
|---|---|---|---|---|---|---|
| Sadness | H | 0 | | | 3-3 | 4-2 |
| Sadness | M | 0 | | 2-2 | 3-1 | |
| Sadness | L | 0 | 1-3 | | | |
| Anger | H | 0 | 1-1 | | 3-3 | |
| Anger | M | 0 | | | 3-3 | |
| Anger | L | 0 | | | | 4-1 |
| Fear | H | 0 | 1-1 | | 3-3 | |
| Fear | M | 0 | 1-1 | | 3-3 | |
| Fear | L | 0 | | 2-2 | | |
| Surprise | H | 0 | | | 3-3 | 4-2 |
| Surprise | M | 0 | 1-1 | | | |
| Surprise | L | 0 | | 2-2 | | |
| Joy | H | 0 | 1-2 | | | |
| Joy | M | 0 | 1-3 | | | |
| Joy | L | 0 | 1-3 | | | |
| Tranquility | H | 0 | 1-1 | 2-2 | | |
| Tranquility | M | 0 | | 2-2 | 3-1 | |
| Tranquility | L | 0 | 1-3 | | | |

**Figure 2 — Social**

| Social | Level | \ | \ | \ | \ | \ |
|---|---|---|---|---|---|---|
| Sadness | H | 0 | 1-1 | | 3-2 | |
| Sadness | M | 0 | | | | 4-1 |
| Sadness | L | 0 | 1-1 | | | |
| Anger | H | 0 | 1-1 | | 3-2 | |
| Anger | M | 0 | 1-3 | 2-1 | | |
| Anger | L | 0 | | | 3-4 | |
| Fear | H | 0 | 1-3 | 2-3 | | |
| Fear | M | 0 | | | 3-1 | |
| Fear | L | 0 | 1-3 | | | |
| Surprise | H | 0 | 1-2 | 3-1 | | |
| Surprise | M | 0 | 1-1 | 2-3 | | |
| Surprise | L | 0 | 1-1 | | 3-4 | |
| Joy | H | 0 | 1-1 | | | |
| Joy | M | 0 | 1-1 | | | |
| Joy | L | 0 | | | 3-4 | |
| Tranquility | H | 0 | 1-1 | | | |
| Tranquility | M | 0 | 1-1 | | 3-4 | |
| Tranquility | L | 0 | 1-1 | | | |

**Figure 2.** Psychological theories and techniques included in the proposed psychological narrative. H = high, M = medium, L = low.

Figures 1 and 2 show outcomes from applying a single psychotherapeutic theory or technique, or mixed theory or technique, considering specific elderly situations and emotional states. Effective psychological therapies for the elderly were selected based on previous studies. The proposed narrations mainly applied positive-psychological techniques for positive emotions, such as joy and tranquility, to help sustain and maintain those states. On the other hand, various theories and techniques were combined for negative emotions, such as sadness and anger, focusing on CBT or ACT. Using single and combined theories and techniques leverages both traditional and postmodern psychotherapy and particularly breaks away from existing fixed methods.

Figure 3 is an example of psychotherapy narration that is three psychotherapeutic theories were applied. First, the pink part is applied PCT. The point to note is the counselor's attitude, that is, congruence, genuineness, positive acceptance, and empathetic understanding. This is the golden rule in the relationship between counselor and client [20]. Second, the green part is applied WBT. The basic concept of WBT helps clients improve their level of psychological well-being by facilitating past positive experiences [23]. Third, the yellow part is applied ACT. Among the techniques of ACT, the purpose of contact with the present moment is to accept the client's present experience and develop psychological flexibility [43]. The complex psychological technique applied in this narration to help the mental well-being of the elderly is based on psychological literature. Along with the previous premise, this study analyzed the level of satisfaction by situation and emotion of the elderly as an initial psychotherapy-narrative model.

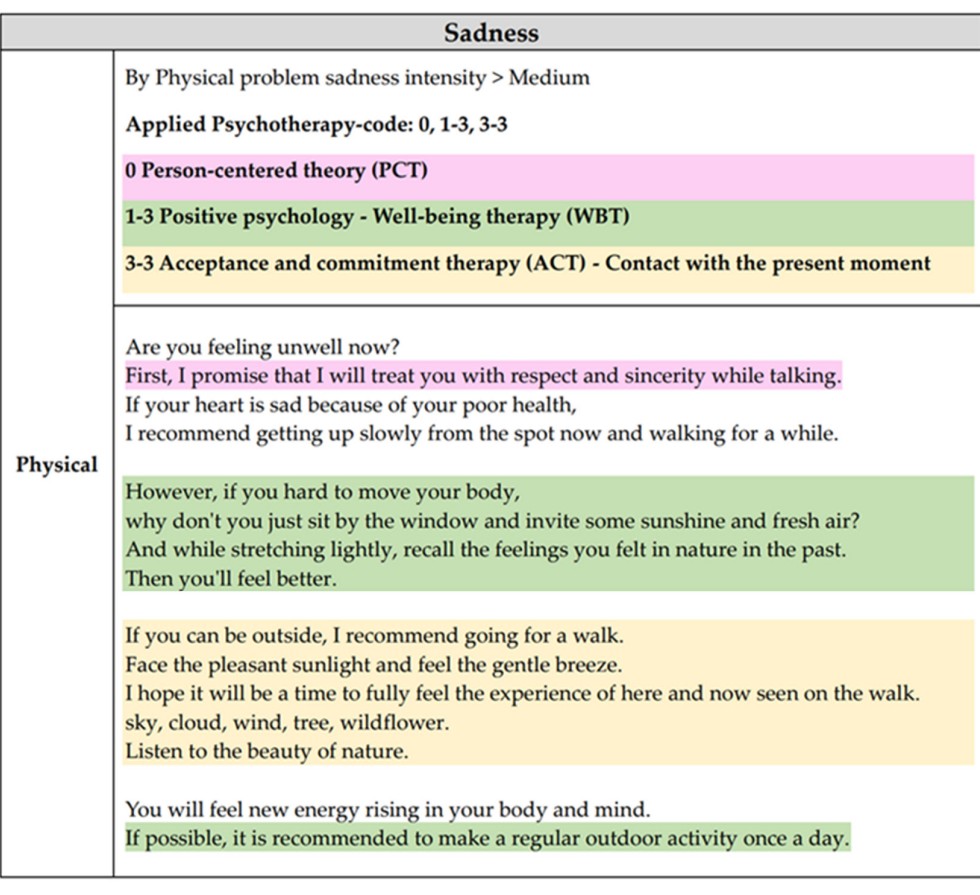

**Figure 3.** An example of psychotherapy narration.

### 3.2. Study Subjects

This study conducted a survey on 100 general elderly people aged 65 years or older. We employed a simple random sampling method to select subjects from those who could read and respond to the questionnaire, understood the study's purpose, and agreed to participate. The survey was conducted from 22 August to 8 September 2022 and received IRB (Institutional Review Board) approval (DUIRB-202208-18). Table 2 summarizes the demographics of the 100 selected participants.

**Table 2.** Survey participant demographics.

|  |  | n | Proportion (%) |
|---|---|---|---|
| Gender | Male | 34 | 34.0 |
|  | Female | 66 | 66.0 |
| Age group | >80 | 4 | 4.0 |
|  | 70–79 | 4 | 4.0 |
|  | 65–69 | 92 | 92.0 |
| Highest academic level | Elementary | 5 | 5.0 |
|  | Middle school | 3 | 3.0 |
|  | High school | 32 | 32.0 |
|  | University | 53 | 53.0 |
|  | Graduate school | 7 | 7.0 |
| Major income source | Self-employed | 7 | 7.0 |
|  | Real estate | 15 | 15.0 |
|  | Pension | 39 | 39.0 |
|  | Family members (e.g., children) | 35 | 35.0 |
|  | Other | 4 | 4.0 |

### 3.3. Study Instrument and Data Collection

Each questionnaire item in this study was modified and supplemented with reference to related questionnaires from previous studies. After completion, each questionnaire was verified by two independent experts: a Doctor of Psychology major and an expert practitioner from the psychological counseling research center.

The final questionnaire comprised five demographic questions regarding recent stressful situations in daily life and how to cope with them, with six questions related to emotions felt during the situation. There were 15 questions in total, including 4 questions regarding participant satisfaction with the proposed psychotherapy narration, focusing on mental well-being corresponding to emotion.

### 3.4. Data Analysis

3.4.1. Frequency Analysis

Frequency analysis was used to identify general characteristics for the study subject's demographics and descriptive statistics were derived to indicate overall satisfaction with the psychotherapy narration according to stress situations and emotions.

3.4.2. Chi-Square Test

The chi-square test was employed to verify differences in stress coping methods (stress resolution methods, desired conversation partners) by gender and emotional intensity for stressful situations.

## 4. Results

Table 3 shows that respondents preferred mechanisms to deal with stress. As a result of conducting a chi-square test to verify the difference in stress relief methods between genders, it was found that there was no significant difference at a significant probability of 5%. Walking was identified as the most significant method to relieve negative emotions due to stressful situations for men (44.1%), with the next highest being conversation (20.6%). Women exhibited similar relative outcomes but had slightly higher proportions, with walking as the highest (51.5%) and then conversation (24.2%), i.e., a significantly higher proportion of women preferred conversation to help deal with stressful situations. Thus, 75.7% and 64.7% of women and men, respectively, prefer walks and conversations to relieve negative emotions.

**Table 3.** Preferred mechanisms to deal with stress by gender.

| | Preferred Mechanism to Deal with Stress [Respondent Count and Proportion (%)] | | | | | | | | |
|---|---|---|---|---|---|---|---|---|---|
| | **Walking** | **Eating** | **Drinking or Smoking** | **Reading** | **TV** | **Conversation** | **Shopping** | **Religion** | **Other** |
| Male | 15 (44.1) | - | 4 (11.8) | - | 5 (14.7) | 7 (20.6) | 1 (2.9) | 1 (2.9) | 1 (2.9) |
| Female | 34 (51.5) | 3 (4.5) | - | 1 (1.5) | 4 (6.1) | 16 (24.2) | 5 (7.6) | 2 (3.0) | 1 (1.5) |
| Total | 46 (46.0) | 3 (3.0) | 4 (4.0) | 1 (1.0) | 9 (9.0) | 23 (23.0) | 6 (6.0) | 3 (3.0) | 2 (2.0) |
| | Chi-square = 13.102, *p*-value = 0.108 | | | | | | | | |

Table 4 shows the most recent stress situation experienced by the survey participants. As a result of conducting a chi-square test to verify the difference in the most recent stressful situations between genders, it was found that there was no significant difference at a significance probability of 5%. Among the stressful situations, the largest number of respondents in health answered that they were stressed (58.0%), followed by social, economic, and emotional cognition in that order. These outcomes are consistent with previous studies. In principle, health-related stresses become more common with age, and hence are more likely to have been the most recent stress experienced. More practically, Yang and Oh [6] showed that the most common cause of elderly stress was caused by physical change, and Lee and Lee [49] showed that perceived stress level was the highest for health status.

**Table 4.** Most recent stress situations by gender.

| | Stress Situations [Respondent Count and Proportion (%)] | | | | |
|---|---|---|---|---|---|
| | **Health** | **Social** | **Mental** | **Economy** | **Total** |
| Male | 19 (55.9) | 6 (17.6) | 4 (11.8) | 5 (14.7) | 34 (100.0) |
| Female | 39 (59.1) | 10 (15.2) | 8 (12.1) | 9 (13.6) | 66 (100.0) |
| Total | 58 (58.0) | 16 (16.0) | 12 (12.0) | 14 (14.0) | 100 (100.0) |
| | Chi-square = 0.148, *p*-value = 0.986 | | | | |

Table 5 shows participants' emotions under stressful situations. As a result of conducting a chi-square test to verify the difference in emotions under stress situations between genders, it was found that there was no significant difference at a significance probability of 5%. Women experienced sadness, fear, surprise, and anger in decreasing order, whereas men experienced anxiety, sadness, surprise, and anger.

Table 6 shows mean participant emotion intensities on a five-point scale, where one is the least and five is the most intense, in their most recent stressful situations. The emotional intensity was four for all the highest stressor-emotion cases. Fear was the highest emotion intensity when the most recent stress was health-related, whereas social environment and emotional awareness were sadness and the economy was anxiety.

Table 7 shows participant satisfaction with the psychological narration to help resolve their strongest emotion for the stressful situation on a five-point scale, where one is least satisfied and five is most satisfied.

**Table 5.** Participant emotions under stress situations by gender.

| | Emotions under Stress Situations [Respondent Count and Proportion (%)] | | | | | |
|---|---|---|---|---|---|---|
| | **Sadness** | **Anger** | **Fear** | **Surprise** | **Tranquility** | **Total** |
| | 8 (23.5) | 5 (14.7) | 15 (44.1) | 6 (17.6) | - | 34 (100.0) |
| | 28 (42.4) | 3 (4.5) | 25 (37.9) | 9 (13.6) | 1 (1.5) | 66 (100.0) |
| | 36 (36.0) | 8 (8.0) | 40 (40.0) | 15 (15.0) | 1 (1.0) | 100 (100.0) |
| | Chi-Square: 6.095, *p*-value: 0.192 | | | | | |

**Table 6.** Participant emotion intensity for stressful situations.

| | | n (%) | Intensity [Respondent Count and Proportion (%)] | | | | |
|---|---|---|---|---|---|---|---|
| | | | **1** | **2** | **3** | **4** | **5** |
| Health | Sadness | 18 (31.0) | - | 1 (5.6) | 7 (38.9) | 6 (33.3) | 4 (22.2) |
| | Anger | 2 (3.4) | - | - | - | 1 (50.0) | 1 (50.0) |
| | Fear | 26 (44.8) | - | 4 (15.4) | 8 (30.8) | 11 (42.3) | 3 (11.5) |
| | Surprise | 11 (19.0) | - | - | 6 (54.5) | 3 (27.3) | 2 (18.2) |
| | Tranquility | 1 (1.7) | - | - | 1 (100.0) | - | - |
| Social | Sadness | 9 (56.3) | - | - | 4 (44.4) | 4 (44.4) | 1 (11.1) |
| | Anger | 3 (18.8) | - | - | 1 (33.3) | 2 (66.7) | - |
| | Fear | 1 (6.3) | - | - | - | 1 (100.0) | - |
| | Surprise | 3 (18.8) | - | 1 (33.3) | 2 (66.7) | - | - |
| | Tranquility | - | - | - | - | - | - |
| Mental | Sadness | 7 (58.3) | - | - | 4 (57.1) | 3 (42.9) | - |
| | Anger | 2 (16.7) | - | - | - | 1 (50.0) | 1 (50.0) |
| | Fear | 3 (25.0) | - | - | 1 (33.3) | 2 (66.7) | - |
| | Surprise | - | - | - | - | - | - |
| | Tranquility | - | - | - | - | - | - |
| Economic | Sadness | 2 (14.3) | - | - | 1 (50.0) | 1 (50.0) | - |
| | Anger | 1 (7.1) | - | - | - | 1 (100.0) | - |
| | Fear | 10 (71.4) | 1 (10.0) | 1 (10.0) | 4 (40.0) | 2 (20.0) | 2 (20.0) |
| | Surprise | 1 (7.1) | - | - | - | 1 (100.0) | - |
| | Tranquility | - | - | - | - | - | - |

**Table 7.** Average score of satisfaction according to stressful situation.

| Situation Type | | Emotion Intensity [Respondent Count] | | | | | |
|---|---|---|---|---|---|---|---|
| | | 1 | 2 | 3 | 4 | 5 | Summary |
| Health | score | - | 3.0 | 3.68 | 4.05 | 4.0 | 3.81 |
| | n | - | 5 | 22 | 21 | 10 | 58 |
| Social | score | - | 4.0 | 3.57 | 3.43 | 5.0 | 3.63 |
| | n | - | 1 | 7 | 7 | 1 | 16 |
| Mental | score | - | - | 2.8 | 4.17 | 3.0 | 3.5 |
| | n | - | - | 5 | 6 | 1 | 12 |
| Economic | score | 4.0 | 5.0 | 3.4 | 3.6 | 3.0 | 3.57 |
| | n | 1 | 1 | 5 | 5 | 2 | 14 |
| Summary | score | 4.0 | 3.43 | 3.51 | 3.90 | 3.86 | 3.71 |
| | n | 1 | 7 | 39 | 39 | 14 | 100 |

Participants read the psychotherapy narration corresponding to their emotions according to the stressful situation. Satisfaction with whether negative emotions were resolved was measured on a five-point scale. The elderly who experienced stress concerning their health had the highest satisfaction score (average score = 3.81) for the relevant narration.

Table 8 shows participant satisfaction with the psychological narration according to emotion intensity using a 5-point scale where 1 is least satisfied and 5 is most satisfied. The average satisfaction score with psychotherapy narration was highest for surprises at 3.93, followed by sadness (3.83).

**Table 8.** Average score of satisfaction according to emotions.

| Situation Type | | Emotion Intensity | | | | | |
|---|---|---|---|---|---|---|---|
| | | 1 | 2 | 3 | 4 | 5 | Summary |
| Sadness | score | - | 4.0 | 3.31 | 4.21 | 4.4 | 3.83 |
| | n | - | 1 | 16 | 14 | 5 | 36 |
| Anger | score | - | - | 3.0 | 3.2 | 2.5 | 3.0 |
| | n | - | - | 1 | 5 | 2 | 8 |
| Fear | score | 4.0 | 3.2 | 3.62 | 3.81 | 3.6 | 3.65 |
| | n | 1 | 5 | 13 | 16 | 5 | 40 |
| Surprise | score | - | 4.0 | 3.75 | 4.0 | 4.5 | 3.93 |
| | n | - | 1 | 8 | 4 | 2 | 15 |
| Tranquility | score | - | - | 4.0 | - | - | 4.0 |
| | n | - | - | 1 | - | - | 1 |
| Summary | score | 4.0 | 3.43 | 3.51 | 3.90 | 3.86 | 3.71 |
| | n | 1 | 7 | 39 | 39 | 14 | 100 |

## 5. Discussion

The National Statistical Office (2022) showed that 65 and older accounted for 16.8% of the total South Korean population, i.e., approximately one in six people, representing an approximately 5.5% increase since 2015 (11.3%) [4]. They define 7% or more aged 65 or older as an aging society, 14% or more as an aged society, and 20% or more as a post-aged society. South Korea is becoming an "aged society," and is expected to reach the "post-aged" level by 2026 (approximately 20.8%). This population aging has exposed several emerging social problems closely related to the quality of life of the elderly [50]. There is a need for psychological treatment to cope with these rapid social changes and the rapid increase in the emotional problems of the elderly.

This study developed a psychological narration for treatments to help the elderly with their mental well-being by emphasizing the importance of coping with stress and managing emotions to improve their quality of life. We conducted a satisfaction survey for the proposed narrations to identify stressful situations experienced by the elderly and their corresponding emotions and subsequently assess the proposed narration's effectiveness. Survey participants included 100 elderly (65 years or older), and results were analyzed using SPSS, with the following conclusions.

First, we selected 100 survey participants, 34.0% male and 66.0% female, and 92% aged 65–69. University graduates accounted for 53.0% and high school graduates 32.0%. Their most common main source of income was the pension (39.0%), with family members (e.g., children) next (35.0%).

Second, more than half of the respondents' (58%) most recent stressor was health-related, then social environment (16%), economy (14%), and emotions (12%), which is consistent with previous studies that represented the perceived health problems of the elderly as their main cause of stress [6,49]. In this regard, fear was the most generated emotion (40%), then sadness (36%), surprise (15%), anger (8%), and tranquility (1%). Thus, more than 70% felt anxious or sad about stressful situations, and many elderly experiences stress accompanied by negative emotions, such as sadness and depression, due to health problems (e.g., reduced physical function and physical strength).

Third, survey participants indicated that they preferred to take a walk to relieve negative emotions due to stressful situations (46%) or talk to other people (23%). Through the above survey, it was possible to examine the status of the elderly's demographic information, major stress situations, major emotions caused by stress, and ways to cope with stress. This information can be the basis for developing customized narration in the future. However, there are limitations as the information is only made in a specific area of Korea.

In another study, they were asked to follow the narration and record their experience regarding resolving negative emotions and their level of satisfaction with the supplied narration (5-point scale, 1 = least satisfied, 5 = most satisfied). Health concerns caused the greatest amount of stress, with sadness being the most intense emotion, and narration satisfaction was the highest for sadness under health-related stress. Participants confirmed that the proposed psychotherapy narration provided high emotional satisfaction. The proposed narration, incorporating relevant PCT, positive psychology, and CBT aspects, can help relieve negative emotions and improve elderly mental well-being by preceding research. However, in this study, only the elderly's level of satisfaction with the narration was tested, and further studies on its detailed effectiveness are needed.

This study has some limitations that should be considered and addressed in future research.

First, the proposed psychotherapy narration can be used alone anytime and anywhere, but there is no interaction, i.e., counselor and client exchanging opinions—a common counseling method. Although the psychotherapy narration currently has no interaction, future development of AI-based psychotherapy narration will allow counselors and clients to communicate directly.

Second, this was a stand-alone study, and hence so it is difficult to assess the proposed psychotherapy-narration continuity and effectiveness. Therefore, continued development and research are important for the improvement of this model.

Third, this study does not include a specific control group (hence the difficulty in deriving wide-ranging results). Future psychotherapy-narrative studies will expand this work and develop more systematic and accurate analyses, including multi-session studies and appropriate control group(s).

## 6. Conclusions

Emotional management for stressful situations is essential to improve the quality of life of the elderly. Psychotherapy narration can be used at any time in a comfortable

place and remotely, and hence offers a useful emotional management tool for the elderly. This study separated common elderly situations and emotions and organized a systematic psychotherapy narration. The proposed psychotherapy narrative embodies postmodern psychotherapy by applying various techniques. It is expected that future postmodern psychotherapy will break away from current psychotherapeutic methods and configure customized psychotherapy suitable for elderly individuals and their specific situations. Through the combination of this accumulated information and AI technology, it is possible to search for and recommend an appropriate psychotherapeutic configuration for a specific client. In the future, as the psychotherapy narrative of this study continues to be developed, it is certain that it will be valuable for the mental well-being of the elderly.

**Author Contributions:** Conceptualization, E.K., S.K. and J.R.; Data curation, E.K.; Funding acquisition, J.R.; Investigation, E.K. and S.K.; Methodology, S.K. and J.R.; Project administration, E.K. and J.R.; Resources, J.R.; Supervision, J.R.; Validation, E.K., S.K. and J.R.; Writing-original draft, E.K. and S.K.; Writing-review and editing, E.K., S.K. and J.R. All authors have read and agreed to the published version of the manuscript.

**Funding:** The Institute of Information & communications Technology Planning & Evaluation (IITP) of the Korean government (MSIT) funded this research (No. 2021-0-00316, Development and demonstration of 4IR (Industrial Revolution)-based non-face-to-face silver healthcare platform technology for mental health management).

**Institutional Review Board Statement:** We conducted this study following the Declaration of Helsinki and approved by the Institutional Review Board of Dongguk University (approval number DUIRB-202208-18) for human studies.

**Informed Consent Statement:** We obtained informed consent from all subjects involved in the study.

**Data Availability Statement:** The authors can provide data upon request.

**Conflicts of Interest:** The authors declare no conflict of interest.

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
