# Peer review of "A Study on Model of Psychotherapy Narration Focused on Mental Well-Being for Stress Management in the Elderly"

_sustainability, doi:10.3390/su15032656_

Round 1

Reviewer 1 Report

For the authors’ guidance my evaluation and some constructive remarks that would help to improve the paper’s quality are included below:

General Comments: First of all, I would like to congratulate the authors for their efforts. This is a well-structured paper. There are many interesting and important insights in this article. In more detail, these are the main points that could be improved:

Abstract

The abstract is very broad and descriptive. It ought to include specific and analytical information about aim, objective, originality, research findings, methodology, and so on. I strongly recommend an analytical abstract including a summary of research findings and the specific contribution of this research.

Introduction

In the introduction section, I recommend the authors to clarify the main argument and support it with the relevant literature in a more systematic manner. My suggestion would be a clear explanation of the problem as the authors see it. In the current form, the introduction is very descriptive, and argumentation is not in action. The literature review is disconnected and does not support the argumentation process. The statistical indicators of the National Statistical Office can be moved to the Discussion section.

Research Questions: The authors did not specify specific research questions. The authors may add some specific research questions and follow-up questions. In case when specific research questions are identified, then the authors can better attract the attention of readers who are able to conceive the theoretical framework of investigation and the authors’ approaches at first glance. I recommend the authors to come up with some specific research questions and follow-up questions, and why the research is interesting and relevant to the field.

Conceptual and Methodological Clarity: The concepts and descriptors used throughout would benefit from more clarity. There is no justification and further clarification regarding how to properly conduct a survey. I recommend the authors to be more inclusive or specify justification for methodological approach and the limitation of their research.

Literature review

The key arguments would benefit strongly from further fleshing out. The authors can enrich the literature review section through adding some up-to-date and relevant scientific materials. I do not understand why the authors did not examine the forthcoming scientific works. I recommend the authors focus on the literature published in 2022 and forthcoming scientific works for 2023 (Only one article published in 2022 was cited by the authors).

Methodology

I recommend the authors clarifying the methodology in detail (e.g., research paradigm: positivism / post-positivism / constructivism / critical theory, research design, research tools, and so on), making sure that the planned methods/research tools are fully detailed. They ought to give attention to justifying the chosen methodology and source of data in terms of demonstrating applicability, adjustment, and usefulness in the paper.

Originality: I have detected a 6% iThenticate Similarity Index Analysis Score that means the paper is an original work.

Concluding Remarks

My suggestion would be a clear explanation of the problem. I suggest looking at the conclusion’s clear argument around their specific contribution to the research field. The authors ought to clarify the originality, added value, contribution to the existing literature, implications for future studies, and so on.

The study ought to have a stronger focus, compelling argument, general and specific research questions, and impressive discussion, and an indication of why the paper holds value to the readership of the Sustainability (MDPI). I recommend the authors reconsider the approach adopted here; think about the research questions they wish to examine; make sure the literature review is a lot more cohesive, and make sure the link between the research questions and empirical results is a lot “tighter” than presented herewith.

Author Response

Thank you so much for explaining in detail the overall parts of my thesis. 
I tried to correct the thesis according to the improvement direction and recommendation you suggested.

Point 1: [Abstract] 
[I strongly recommend an analytical abstract including a summary of research findings and the specific contribution of this research.]

Response 1: Following your comments above, I added a summary of the research findings and contributions of this study to the end of the abstract.

Point 2: [Introduction] 
(1) [I recommend the authors to clarify the main argument and support it with the relevant literature in a more systematic manner. The literature review is disconnected and does not support the argumentation process.]

Response 2-(1): Following your comments above, I changed the research focus to the Psychotherapy-narration model and its satisfaction rather than effectiveness. Therefore, the argument in this study was supported through several literatures in the narration related to psychological theory.

(2) [The statistical indicators of the National Statistical Office can be moved to the Discussion section.]

Response 2-(2): Following your comments above, I moved the sentences to the Discsussion section. By moving these sentences, the flow of the text seems to be more harmonious.

Point 3: [Research Questions]
[I recommend the authors to come up with some specific research questions and follow-up questions, and why the research is interesting and relevant to the field.]

Response 3: Following your comments above, I added some specific research questions and follow-up questions. I believe that these changes set the thesis to the right direction.

Point 4: [Conceptual and Methodological Clarity]
[There is no justification and further clarification regarding how to properly conduct a survey. I recommend the authors to be more inclusive or specify justification for methodological approach and the limitation of their research.]

Response 4: Following your comments above, I focused to the Psychotherapy-narration model and its satisfaction level in elderly.And I added some of limitation of the research in discussion.

Point 5: [Literature review]
[I recommend the authors focus on the literature published in 2022 and forthcoming scientific works for 2023]

Response 5: Following your comments above, I added 7 literature released in 2021-2023. I agree your opinion that is great academic significance to continue and share the latest information.

Point 6: [Methodology]
[They ought to give attention to justifying the chosen methodology and source of data in terms of demonstrating applicability, adjustment, and usefulness in the paper.]

Response 6: Following your comments above,I added one example of narration applied with psychotherapeutic techniques in methods.

Concluding Remarks

Overall In line with your comments, I have improved the clarity of the argument, the points of contribution, the up-to-date literature, the research question, the broadening of the discussion, the provision of value for the reader, and the connection between the research question and the results.

Your comments really helped me a lot, I really appreciate it. In the future, this will be the basis for my research.

Reviewer 2 Report

The title of the manuscript is very long (25 words). It is recommended to review and limit it, so that its extension does not exceed 15 words maximum. The keywords are adequate and coherent with the research problem.

The introduction states that interest has increased in exploring the conditions that favor a full old age in the world, as a response to the increase in the population of older people in the world and the difficulties they face during this stage of the life cycle. The elderly generally experience physical and cognitive decline, loss of a friend or spouse, reduced social participation, and reduced economic capacity. Therefore, the situations in which the elderly experience stress can be divided into four categories: physical, economic, social and mental [6]. Most of the elderly experience physical difficulties due to aging, and the most common cause of stress in the elderly is due to physical changes, and many elderly experience maladjustment or difficulty accepting psychological and physical changes. They often lose status and social roles due to declining health, creating a vicious cycle that leads to significant psychological problems. Overall, the section adequately contextualizes the research problem and provides updated elements on its approach. However, there is a stereotyped view of aging in which only negative aspects are presented, making invisible the challenges and opportunities that older people also face at this stage of development, some of which may favor the emergence of optimal aging trajectories. It is recommended to incorporate studies that address optimal aging, generativity in the elderly, the meaning of life and its relationship with healthy aging, etc. On the other hand, the elements provided around psychotherapies are too extensive. A greater capacity for synthesis on the part of the authors is recommended to highlight and briefly explain the specific principles and characteristics of each therapeutic model presented.

Regarding the method, adequate information is provided regarding the study design, the sample, the instruments and the data analysis strategy. However, it would be desirable that clearer and more specific elements be provided about the psychometric properties of the instrument built for data collection and the strategy used for its validation.

Regarding the results, walking was identified as the most significant method to alleviate negative emotions in stressful situations for men (44.1%), the next highest being conversation (20.6%). Women exhibited similar relative results, but slightly higher proportions, with more walking (51.5%) and talking (24.2%) as strategies used to deal with stressful situations. The tables are adequate, although they are explained in a very superficial way. It is recommended to improve these observations.

On the other hand, the discussion is poorly and superficially constructed. That is, in the section they reaffirm some of their results, but they do not contrast them with a new review of the state of the art, seeking a deeper and more reflective analysis of their findings. Likewise, the limitations are described in a precarious way, so it is recommended to review and structurally correct the entire section. Nor do they provide projections of the study or manage to highlight the scientific novelty offered by this research.

The manuscript does not present a conclusions section. Check and correct.

Regarding the references, a detailed and exhaustive review of the same is recommended, ensuring proper compliance with the editorial standards of the journal.

Author Response

Thank you so much for explaining in detail the overall parts of my thesis. 
I tried to correct the thesis according to the improvement direction and recommendation you suggested.

Point 1: [Title] 
[The title of the manuscript is very long (25 words). It is recommended to review and limit it, so that its extension does not exceed 15 words maximum.]

Response 1: Following your comments above, The number of words in the title of this thesis was changed to 15 words.

Point 2: [Introduction]

[It is recommended to incorporate studies that address optimal aging, generativity in the elderly, the meaning of life and its relationship with healthy aging, etc.]

Response 2-1: Following your comments above, I added this [These programs or environments lead to successful aging of the elderly. In a study by Hwang and Jung [52], the elderly's self-efficacy and social participation activities were closely related to successful aging. Therefore, to achieve optimal aging of the elderly, it is important to understand the detailed emotional aspects and situations for mental wellbeing of elderly. ]

[On the other hand, the elements provided around psychotherapies are too extensive. A greater capacity for synthesis on the part of the authors is recommended to highlight and briefly explain the specific principles and characteristics of each therapeutic model presented.]

Response 2-2: Following your comments above, I changed my title and I focused to the Psychotherapy-narration model and its satisfaction level in elderly.

Point 3: [Regarding the method]
[However, it would be desirable that clearer and more specific elements be provided about the psychometric properties of the instrument built for data collection and the strategy used for its validation.]

Response 3: Following your comments above, I added one example of narration applied with psychotherapeutic techniques in methods.

Point 4: [Discussion]
[Nor do they provide projections of the study or manage to highlight the scientific novelty offered by this research.]

Response 4: Following your comments above, A few argumens about the value and future potential of this study have been added in discussion and conclusion.

Point 5: [Conclusion]
[The manuscript does not present a conclusions section. Check and correct.]

Response 5: Following your comments above, I added a concluding section.

Your comments really helped me a lot, I really appreciate it. In the future, this will be the basis for my research.

Reviewer 3 Report

Study topic is interesting, but design and statistical analysis should be reconsidered to assess the effectiveness of this program.

Author Response

Thank you so much for explaining in detail the overall parts of my thesis. 
I tried to correct the thesis according to the improvement direction and recommendation you suggested.

Point 1: [About Effectiveness]
[Study topic is interesting, but design and statistical analysis should be reconsidered to assess the effectiveness of this program.]

Response 1: Following your comments above, I restructured the title and research question by focusing on satisfaction rather than effetiveness.

Your comments really helped me a lot, I really appreciate it. In the future, this will be the basis for my research.

Round 2

Reviewer 1 Report

I wish to express my sincere appreciation to you for your scientific efforts during the revision process. Great job. Congratulations!

Author Response

Thanks to reviewer 1, it seems that a scientific and systematic framework was formed. And I supplemented the contents of the thesis a little.

Thank you very much.

Reviewer 2 Report

The work has been improved in its various components, integrating the contributions and recommendations of the reviewers.

Despite the obvious shortcomings observed in the method and discussion, the theme, scope and projections that may emerge from the study carried out are valued.

Author Response

Thanks to reviewer 2, I was able to strengthen the characteristics of various parts of the thesis. Obviously, there has been only very basic research and development so far, we will strive for more advanced research to meet the expectations of you.

Thank you very much.  

Reviewer 3 Report

This manuscript was improved after correction in accordance with reviewers' comments. I think it's suitable for publication in the journal.

Author Response

It seems that the thesis has grown further with the opinions of reviewer 3. And I supplemented the contents of the thesis a little.

Thank you very much.
